# A Proteomic Investigation to Discover Candidate Proteins Involved in Novel Mechanisms of 5-Fluorouracil Resistance in Colorectal Cancer

**DOI:** 10.3390/cells13040342

**Published:** 2024-02-14

**Authors:** Mario Ortega Duran, Sadr ul Shaheed, Christopher W. Sutton, Steven D. Shnyder

**Affiliations:** 1Institute of Cancer Therapeutics, University of Bradford, Bradford BD7 1DP, UK; marioortegaduran@gmail.com (M.O.D.); c.w.sutton@Bradford.ac.uk (C.W.S.); 2Nuffield Department of Surgical Sciences, John Radcliffe Hospital, University of Oxford, Oxford OX3 9BQ, UK; sadr.shaheed@nds.ox.ac.uk

**Keywords:** colorectal cancer, drug resistance mechanisms, in vitro models, proteomics, 5-fluorouracil, stable isotope labelling with amino acids in cell culture (SILAC)

## Abstract

One of the main obstacles to therapeutic success in colorectal cancer (CRC) is the development of acquired resistance to treatment with drugs such as 5-fluorouracil (5-FU). Whilst some resistance mechanisms are well known, it is clear from the stasis in therapy success rate that much is still unknown. Here, a proteomics approach is taken towards identification of candidate proteins using 5-FU-resistant sublines of human CRC cell lines generated in house. Using a multiplexed stable isotope labelling with amino acids in cell culture (SILAC) strategy, 5-FU-resistant and equivalently passaged sensitive cell lines were compared to parent cell lines by growing in Heavy medium with 2D liquid chromatography and Orbitrap Fusion™ Tribrid™ Mass Spectrometry analysis. Among 3003 commonly quantified proteins, six (CD44, APP, NAGLU, CORO7, AGR2, PLSCR1) were found up-regulated, and six (VPS45, RBMS2, RIOK1, RAP1GDS1, POLR3D, CD55) down-regulated. A total of 11 of the 12 proteins have a known association with drug resistance mechanisms or role in CRC oncogenesis. Validation through immunodetection techniques confirmed high expression of CD44 and CD63, two known drug resistance mediators with elevated proteomics expression results. The information revealed by the sensitivity of this method warrants it as an important tool for elaborating the complexity of acquired drug resistance in CRC.

## 1. Introduction

5-Fluorouracil (5-FU)-based chemotherapy has been used during the last 60 years as the main postoperative adjuvant treatment for advanced colorectal cancer (CRC) patients. 5-FU is converted by different routes into several active metabolites in mammalian cells, leading to the inhibition of deoxythymidine monophosphate (dTMP) production: an essential element for DNA repair and replication processes. This leads to cytotoxicity and cell death [1,2].

However, the overall response in advanced CRC patients when they are treated with 5-FU alone is not higher than 15% due to development of mechanisms of resistance [3]. Some of the mechanisms of resistance to 5-FU are well known and can be mitigated for, such as those related to increased levels of the intermediate metabolites thymidylate synthase (TS) and dihydropyrimidine dehydrogenase (DPD), its interactions as a substrate for multidrug resistance cell membrane pumps, such as MDR1 [4], and alterations in cell cycle kinetics or apoptotic pathways. However, it is clear from the stasis in improvement in therapy success rate over the past three decades that there is still much that is undiscovered [5], with other cell- or vesicle-surface proteins possibly having a role in drug resistance, for example CD44 [6,7,8].

Previous studies focused on the discovery of mechanisms of resistance have taken a genomics approach. However, somatic alterations observed between sensitive and resistant tumour cells are not always significant at the transcriptional level [9,10]. Genomics is complemented and enhanced by proteomic analysis, which focuses on global protein changes within a biological system at a specified time or under a set of conditions, thus providing information on functionality in biological and molecular processes [11]. In this respect, a stable isotope labelling with amino acids in cell culture (SILAC)-based method for quantitative proteomic analysis of CRC-resistant sublines to 5-FU is a powerful tool to identify new biomarkers and mechanisms of resistance in vitro, which can be extrapolated to other cancer- and drug-resistance discoveries [12]. The SILAC approach is a simple and straightforward one based on metabolic stable isotope labelling of proteins by growing cells in a medium containing ^13^C and ^15^N lysine and arginine essential amino acids [10,13]. After six to nine cell divisions, all instances of the specific amino acid will be replaced by its isotopic analogue, giving rise to Heavy isotopic proteins [14]. Subsequently, the comparison of two cell lines (e.g., control and experimentally manipulated) with proteins from the non-labelled (Light) control and the labelled (Heavy) experimentally manipulated extract can be combined, digested, and then analysed using LC-MS/MS. The resulting mass spectrometry data facilitate both protein identification and relative quantification [15].

In this study, we employed the SILAC proteomics approach to uncover candidate proteins potentially involved in novel mechanisms of 5-FU resistance within internally generated drug-resistant CRC cell lines. Our focus on selected proteins revealed subtle patterns of differential expression, providing key insights into the intricacies of 5-FU resistance. Subsequent validation through advanced immunodetection methods further reinforced these findings, contributing to compelling evidence for investigating these candidate proteins in future studies for their involvement in novel mechanisms of drug resistance in CRC.

## 2. Materials and Methods

### 2.1. Materials

Two human CRC cell lines, DLD-1 and HT-29, were selected for this study due to their sensitivity to 5-FU in preliminary in-house studies. A mammary adenocarcinoma cell line, MDA-MB-231, was included as a CD44-expressing cell line for validation studies. The human DLD-1 colon adenocarcinoma cell line was obtained from ATCC (LGC Standards, Middlesex, UK) whilst the HT-29 human colon adenocarcinoma and MDA-MB-231 human mammary adenocarcinoma cell lines were obtained from the National Cancer Institute Department of Cancer Treatment and Diagnosis Tumour Repository (Frederick, MD, USA). The cells were maintained in an RPMI 1640 culture medium supplemented with 10% (*v*/*v*) foetal bovine serum (FBS), 1 mmol/L sodium pyruvate, 2 mmol/L of L-glutamine (all from Merck, Gillingham, UK) and incubated at 37 °C in 5% CO_2_. Dulbecco’s phosphate-buffered saline (PBS) was used for washing steps.

5-FU (Merck, Gillingham, UK) was prepared as a 11 mM stock solution in dimethyl sulfoxide (DMSO, Merck, Gillingham, UK) and aliquots stored at −20 °C until use.

For the SILAC labelling procedure, stable isotope-labelled amino acids, L-Arginine-HCl (^13^C_6_, ^15^N_4_) and L-Lysine-2HCl (^13^C_6_, ^15^N_2_), 10% (*v*/*v*) dialyzed FBS, and L-Proline were all purchased from Thermo Fisher Scientific (Loughborough, UK). L-lysine and L-arginine, essential amino acids with metabolic incorporation, were chosen for SILAC experiments because they lack biosynthetic pathways in mammalian cells and hence cannot be synthesized de novo, ensuring any signal detected is due solely to incorporation for the experiment [16]. In addition, these amino acids possess properties that make them compatible with mass spectrometric analysis for precise quantification, with the labelled forms readily distinguishable from their unlabelled counterparts. Details of other materials used for proteomics procedures, immunoblotting, and immunofluorescence are given below.

### 2.2. Chemosensitivity Assay

Chemosensitivity was evaluated using the MTT assay as described previously [17]. Briefly, 180 μL of 1 × 10^4^ cells/mL suspension was added to each test well of a 96-well plate and incubated overnight at 37 °C. 5-FU or control solutions were added to each well and the plates cultured under standard conditions for 4 days, after which cells were incubated with MTT solution (5 mg/mL) in PBS for 4 h. Formazan crystals were then solubilised in 150 µL of DMSO and the plates scanned at 540 nm on a Thermo Scientific Multiskan EX plate reader (Thermo Fisher Scientific, Loughborough, UK). Chemosensitivity in terms of the half maximal inhibitory concentration (IC_50_) was then determined from the data.

### 2.3. Development of 5-FU-Resistant Human CRC Sublines

CRC sublines with resistance to 5-FU were established and derived from DLD-1 and HT-29 parent cell lines by continuous exposure to increasing concentrations of 5-FU over a period of ten months (Figure 1). IC_75_ values were used as initial starting doses for DLD-1 and HT-29 parent cell lines, 15 and 18 μM, respectively. 5-FU concentration was gradually increased up to 250 μM in the DLD-1/5-FU cell line and up to 60 μM in the HT-29/5-FU cell line (Figure 1). For each parent cell line, two controls grown in drug-free media were harvested in parallel for further analyses at low (DLD-1 P9 and HT-29 P9) and high (DLD-1 P65 and HT-29 P57) passages.

### 2.4. SILAC Approach for Quantitative Proteomic Analysis

Quantitative proteomics characterization of resistant cell lines was carried out using a stable isotope labelling amino acids in cell culture (SILAC)-based approach. DLD-1 and HT-29 CRC parent cell lines were SILAC labelled with L-Arginine-HCl (^13^C_6_, ^15^N_4_) and L-Lysine-2HCl (^13^C_6_, ^15^N_2_) by culturing them for 9 passages in RPMI Media with 10% (*v*/*v*) dialyzed FBS for SILAC and incubated at 37 °C in 5% CO_2_ (Figure 1). L-Proline was added to avoid the metabolic conversion of Heavy arginine to Heavy proline [18]. The medium was replaced every 4 days once cells reached 65–75% confluency. The SILAC-labelled cell line samples were evaluated in triplicate by assessment of SILAC ratios between unlabelled and labelled versions of lysine-containing peptides and arginine-containing peptides to determine Heavy arginine and lysine incorporation into proteins, respectively. To determine Heavy SILAC-labelling efficiency, protein was extracted from wild-type cells after 9 passages and subjected to Mudpit proteomics (or Shotgun proteomics). In both cases, DLD-1 and HT-29 cell lines grown in SILAC media achieved >98% incorporation of labelled amino acid into their proteins, in line with rates achieved by previous groups for this methodology [19,20].

### 2.5. Protein Extraction and Protein Digestion

All materials for protein extraction buffer were provided from Merck (Gillingham, UK) unless otherwise stated. For each cell line, 1 × 10^7^ cells were harvested, washed with PBS, and proteins were extracted from the cells using protein extraction buffer containing 7 M urea, 2 M thiourea, 0.4% CHAPS, 0.1% sodium dodecyl sulphate (SDS), 0.05% Sodium deoxycholate, and protease inhibitor cocktail from Roche (Welwyn Garden City, UK) in PBS at 4 °C, and then homogenized by sonication with a SH70G Sonicator from Philip Harris Scientific (Lichfield, UK). Samples were centrifuged at 13,000 rpm for 25 min at 4 °C, with the protein yield measured using a Bradford assay kit (Thermo Fisher, Loughborough, UK). Proteomics analysis was performed on paired Light- and Heavy-labelled protein extracts (see Figure 1) in parallel for DLD-1 and HT-29 cell lines. Paired extracts of Light samples (200 µg of protein from Parent cell line, low or high passage control, or resistant sublines) and Heavy samples (200 µg of protein from Parent cell line grown in SILAC media) were combined for protein digestion. The combined extract samples were treated with 1 mL of chilled acetone overnight at −20 °C for protein. Precipitated protein was resuspended in 40 µL of 8 M urea, reduced with 50 mM dithiothreitol (DTT) (Merck, Gillingham, UK) at 70 °C for 15 min, and alkylated with 100 mM iodoacetamide (IAA) (Merck, Gillingham, UK) at room temperature for 15 min. The protein sample was diluted 4-fold and digested overnight at 37 °C, with 1 mg/mL Pierce^TM^ trypsin (Thermo Fisher Scientific, Loughborough, UK) at a protein—protease ratio of 10:1.

### 2.6. Shotgun Proteomics 

Digested peptides from each paired sample underwent strong cationic exchange chromatography (SCX) using an ISOLUTE^®^ SCX column (Biotage, Hengoed, UK). The column was equilibrated with a loading buffer (LB) of 10mM potassium di-hydrogen phosphate (KH_2_PO_4_), 0.01% Sodium azide (pH = 3.0) in 25% acetonitrile (ACN) from Merck (Gillingham, UK), and peptides were eluted and collected sequentially with incremental increases in concentration (30 mM to 1000 mM) of Potassium Chloride (KCl) prepared in loading buffer. A total of 12 SCX fractions were collected and desalted using C18 columns (Kenesis, Cambridgeshire, UK) before lyophilisation using 45 °C aqueous mode of Genevac Centrifugal Evaporator EZ-2 SP (Thermo Fisher Scientific, Loughborough, UK).

All SCX fractions were separated on an Ultimate 3000 HPLC connected on-line to an Orbitrap Fusion™ Tribrid™ Mass Spectrometer (Thermo Fisher Scientific, Loughborough, UK). Each SCX fraction was resuspended in 30 μL of 0.1% formic acid and 10μL of each fraction was loaded on a C18 trap column (300 μm × 5 mm, 100 A) (Thermo Fisher Scientific, Loughborough, UK) at a flow rate 10µL/min. After washing for 4 min with loading buffer A (2% ACN, 0.1% formic acid), peptides were then transferred to a C18 analytical column (75 μm × 50 cm, 2 μm, 100 A) (Thermo Fisher Scientific, Loughborough, UK) at a temperature of 40 °C. The peptides were eluted using a gradient of Solvent B (100% ACN, 0.1% FA): 0 min at 5%, 5 min at 7%, 65 min at 25%, 80 min at 45%, and 85 min at 85%. The eluted peptides were ionized by electrospray ionization (2000 V) using a steel emitter and acquired with an Orbitrap Fusion mass spectrometer coupled with the Nanospray Flex™ Ion Source (Thermo Fisher Scientific, Loughborough, UK), and ion transfer tube temperature was set at 275 °C.

The Orbitrap Fusion mass spectrometer was operated in the data-dependent acquisition (DDA) mode. The MS1 survey scan of parent ion was set from 350 to 1500 *m*/*z*, and data were acquired at a high resolution of 120,000 (*m*/*z* 200); the AGC target was set to 3 × 10^5^ and the maximum injection time was 100ms. The second stage of mass spectrometry (MS2) scans was performed on IonTrap at rapid scan rate, with dynamic exclusion (±5 ppm), 50 s; cycle time, 3 s; isolation width, 0.7 *m*/*z*; and MIPS mode as peptide. Ions with charge states 2–7 were sequentially fragmented by collision-induced dissociation (CID) with a fixed collision energy of 35%. All LC-MS acquisitions were performed in positive ion mode only.

MS/MS spectra generated were analysed in the Mascot 2.4 Database Manager search algorithm (Matrix Science, London, UK) through Proteome Discoverer platform (version 2.2; Thermo Fisher Scientific, Loughborough, UK) using the human database from UniProt (SwissProt Version 2.5.1) with records of 551,705 functional proteins. Search parameters, including up to 2 missed trypsin cleavages, MS1 mass tolerance of 10 ppm, and MS2 mass tolerance of 0.6 Da, were selected. Dynamic modifications included Oxidation (M) and Deamidated (NQ) and Carbamidomethyl (C) as a static modification. A protein identification confidence threshold of *p* < 0.05 with at least one unique peptide and two peptide spectral matches (PSMs) was used to establish lists of quantified proteins. The peptide and protein SILAC ratios were determined using SILAC peptide pair searches with fixed L-Arginine-HCl (^13^C_6_, ^15^N_4_ plus 10Da) and L-Lysine-2HCl (^13^C_6_, ^15^N_2_ plus 8Da) modifications. The mass spectrometry proteomics data have been deposited to the ProteomeXchange Consortium via the PRIDE (https://www.proteomexchange.org/) partner repository with the dataset identifier PXD047672 (data submitted 10 December 2023).

### 2.7. Immunoblotting

Validation of CD44 SILAC proteomic quantification results was carried out by immunoblotting. A sample of 20 µg of whole-cell lysate from each cell line was separated by SDS-PAGE electrophoresis in 12% Acrylamide gels using a Llaemli buffer system flowed by electroblotting onto Hybond-P nitrocellulose blotting membrane (0.45 μm) (Thermo Fisher Scientific, Loughborough, UK) using constant Amperage of 300 mA for 2 h. Polyclonal rabbit Anti-human CD44 (Abcam, Cambridge, UK) at 1:2000 dilution, or mouse Anti-β-Actin at 1:7000 dilution (Merck, Gillingham, UK), was used to verify proteomic analyses. Horseradish peroxidase-conjugated Goat Anti-rabbit IgG (ab6721, Abcam, Cambridge, UK) was used as a secondary antibody. All immunoblots were analysed using GelAnalyzer (http://www.gelanalyzer.com/?i=1, accessed on 10 March 2018) and normalised with respect to β-actin. The MDA-MB-231 mammary adenocarcinoma cell line was used as a positive control cell line for high CD44 expression [21].

### 2.8. Immunofluorescence 

CD63 protein expression was assessed in the DLD-1 parent cell line and DLD-1/5-FU-resistant subline. The DLD-1 parent cell line and DLD-1/5-FU-resistant subline were added at 1 × 10^4^ cells/mL to sterile 22 × 22 mm coverslips in a 6-well plate and incubated overnight at 37 °C. Medium was removed and each well was washed three times with Phosphate-buffered saline (PBS) provided by Severn Biotech (Kidderminster, UK) for 2 min. After PBS washing, 1ml of pre-chilled methanol was used to fix the cells in the freezer for 10 min. After fixing, TPBS (50 μL Triton X-100 from Merck (Gillingham, UK) in 50 mL of PBS) was used to wash each well three times. Blocking for non-specific binding of antibodies was carried out by incubating cells for 1 h at room temperature with 1.5% normal rabbit serum (NRS; Abcam, Cambridge, UK) prepared in TPBS (30 μL NRS + 1970 μL of TPBS). CD63 was detected with the primary mouse monoclonal antibody [TS63] to CD63 (Abcam, Cambridge, UK) overnight, followed by incubation in TRITC-conjugated rabbit anti-mouse secondary antibody (Abcam, Cambridge, UK). Cells were washed, counterstained with DAPI, and mounted with Vectashield^®^ Mounting Media with DAPI (Vector Laboratories, Peterborough, UK) before being subjected to microscopy. Fluorescent images were captured at X40 objective lens magnification using a Leica DM2000 microscope (Wetzlar, Germany). Finally, fluorescence intensities were determined using the Leica Application Suite software v4.0 for quantitative analysis.

### 2.9. Statistical Analysis

All statistical tests for MTT assays and growth curves were generated using GraphPad Prism 5.0 (GraphPad Software, Inc., San Diego, CA, USA). The effect of each drug on the viability of CRC cell lines was measured by MTT assay. Cells in log phase were exposed to indicated concentrations of 5-FU and, after 4 days incubation, cell survival was determined by MTT assay. Results are expressed as the means ± SD of 3 independently repeated experiments. One-way ANOVA statistical analyses, * *p* ≤ 0.05, ** *p* ≤ 0.01, *** *p* ≤ 0.001, and **** *p* ≤ 0.0001, were selected to compare the sensitivity (IC_50_ value) among sublines for statistical analysis (* *p* ≤ 0.05). In all tests, *p* ≤ 0.05 was considered to indicate a statistically significant difference.

The protocol used for data processing in proteomics analysis was as follows. Only master proteins with at least one unique peptide, identified twice in the LC-MS analysis with a Mascot score > 20 (*p* < 0.05) and quantified in both Light and Heavy samples, were selected for downstream analysis. These multi-SILAC datasets were then subjected to analysis using the LIMMA package [22] in R programming. The raw protein abundances for each dataset were normalised using the Median approach and subsequently transformed into log_2_ values to calculate expression differences between groups. Multi-SILAC datasets were generated from three individual SILAC experiments, encompassing a resistant subline, a low-generation control, and a long-generation control. A fold change and its associated *p*-value, with adjusted *p*-values of 5% FDR (Benjamini–Hochberg method) for multiple comparisons, were calculated for each protein according to their significant different ratio expressions between the resistant subline and its two related control cell lines with a high and a low number of passages. All commonly quantified proteins in resistant and parent cell lines (multi-SILAC dataset) were classified into three groups in terms of expression, relative among single SILAC experiments. Three groups were defined as “not altered proteins”, “up-regulated proteins”, and “down-regulated proteins”, using ±1 as log_2_ fold change threshold for up-regulated and down-regulated proteins. The data for all proteins were summarised and plotted as log_2_ fold change versus the −Log10 of the adjusted p-value of LIMMA-modelled proteomics data. Venn diagrams were created with web tool provided by the University of Ghent Belgium, through the Bioinformatics and Evolutionary Genomics department (https://bioinformatics.psb.ugent.be/webtools/Venn/; accessed on 5th May 2018). Protein groups derived from this cluster analysis were subject to gene ontological examination using EnrichNet [23], STRING [24], and DAVID [25] bioinformatics (all accessed on 4 December 2023): web applications to identify significant biological processes, pathways, and functions linked to the resistance mechanism.

## 3. Results

### 3.1. Establishment of DLD-1 and HT-29 Drug-Resistant Sublines to 5-FU

Initial chemosensitivity assays established IC_75_ values for 5-FU of 15 μM for DLD-1 and 19 μM for HT-29, and these concentrations were selected to initiate resistant subline development. The resistance indices were evaluated by MTT according to the relative resistance for pre-treated cell lines with 5-FU in comparison with parent cell lines. This was defined as the ratio of IC_50_ of the resistant subline to the IC_50_ of the sensitive parent cell line. Both cell lines showed a progressive response when grown under increasing 5-FU concentrations and, after ten months of episodic drug exposure, DLD-1/5-FU [IC_50_ = 250 μM] and HT-29/5-FU [IC_50_ = 60 μM]-resistant sublines were established (Figure 2A,B), showing a 130.2-fold and 3.5-fold change in 5-FU resistance, respectively.

### 3.2. Identification of Proteome Changes in DLD-1- and HT-29-Resistant Sublines to 5-FU Using a Mass Spectrometry by SILAC Approach

From the three SILAC experiments for each cell line (resistance and two passage controls), an average of 3640 proteins were quantified (Mascot score ≥ 28, PSMs ≥ 2, peptide SILAC ratios ≥ 2, and at least one unique peptide) in DLD-1 and HT-29 cell lines. There were 3748 proteins common between the three DLD-1 experiments and 4161 common between the HT-29 datasets (Figure 2C,D). Unique proteins in the passage controls were generally low scoring and not detected specifically due to passage related changes. For both cell lines the resistance subtype generated the most unique protein identifications.

Overall, 4622 and 4529 unique gene products were quantified for the DLD-1 and HT-29 cell lines, of which 3003 were common. Analysis of quantified data (raw data deposited 10 December 2023 at https://www.proteomexchange.org/ with the dataset identifier PXD047672) was performed by LIMMA modelling through comparing the relative log_2_-fold change in expression of the resistant cell lines to the short and long generation controls (*p* < 0.05). Dispersion of log_2_-fold change in DLD-1/5-FU proteins was significantly higher (*p* < 0.0.5; *p* = 0.0035) than log_2_-fold change dispersion in HT-29/5-FU proteins. Based on a ±1 log_2_-fold change threshold in the resistant cell lines compared to generation controls, 118 proteins were increased and 93 were decreased in expression in the DLD-1 dataset (Appendix A), with 66 increased and 109 decreased in expression in the HT-29 dataset (Appendix A).

To relate proteins with altered expressed with function, proteins were classified according to five biological processes which may mediate drug response and drug resistance: (1) apoptotic process, (2) DNA repair process, (3) metabolism of drugs and small molecules, (4) intracellular protein transport, and (5) cellular membrane transport and cell membrane organization processes. This is based on molecular functions identified by GO and pathway analysis for selected proteins using EnrichNet.

Of the up-regulated proteins in DLD-1/5-FU, those involved in protein transport (SEC24D, VPS16, VPS36, SYNRG, STAM2, GOPC, TMED4, CORO7, RAB2B, TBC1D17, and CCDC53) were also linked with vesicle formation and trafficking, along with additional up-regulated genes (SEC24D, SYTL2, WASHC3, HIP1, DYNLT3, DCTN3, SPAST). Furthermore, proteins involved in DNA repair, chromatin remodelling (HDAC3, PMS2, TAOK1, ROMO1, PRKCD), transcription (TRIP4, NFAT5, JUND), anti-apoptosis/apoptosis (NOL3, PAWR, PRKCD), stress (NFAT5, EIF2AK4, APP, STK39), small molecule transport (NFAT), and heparin sulphate/matrix degradation (NAGLU, APP, AGRN, ITGA3), were predominant.

Among the uniquely down-regulated proteins in DLD-1/5-FU-resistant subline, those associated with anti-apoptosis/apoptosis proteins (EMC4, DHCR24, TCTN3), small molecule metabolism (FDXR, DHCR24), vesicle formation (VPS45, VPS52, AP3S1, AP3B1, AP3M1), Five Friends of Methylated CHTOP (5FMC) complex (WDR18, TEX10, PELP1) and DNA repair, chromatin remodelling, and H3K9me3 methylation (PDS5B, H2AFY2, SMARCD2, MYBBP1A, TRIM28, RIOX2) were the most significantly altered. Six enzymes associated with lipid metabolism were down regulated (TAMM41, ACSL4, SPTLC3, DHCR24, FDXR, HMGCS1).

Of the up-regulated proteins in HT-29/5-FU-resistant subline, 35 proteins were described as integral components of the transmembrane region (BET1, OCLN, TFRC, TOR4A, FUT4, LDLR, MAN2A1, EPCAM, DST, REEP4, ADGRE5, HLA-A, PLSCR1, ADPGK, TNFRSF21, METTL2B, ATL2, SPINT1, ALG2, ERLIN1, APLP2, HACD3, CPD, MYOF, GOLIM4, CLN6, PLPP2, FDFT1, SLC12A2, CCDC51, FTH1, GALNT7, APP, QSOX2, DSG2) (*p* < 4.45 × 10^−5^ 4.4 × 10^−5^) and eight of these proteins were also associated with both endoplasmic reticulum and Golgi apparatus (HACD3, BET1, ERLIN1, ATL2, CLN6, DST, FDFT1, REEP4).

Within the down-regulated proteins from HT-29/5-FU, eight proteins were associated with cell–cell adhesions (EFHD2, S100P, SH3GLB1, ANXA1, CAPZA1, CSNK1D, LYPLA2, TAGLN2) (*p* < 4.3 × 10^−4^ 4.3 × 10^−4^).

The next step in the analysis was to identify which proteins demonstrated a similar up- or down-regulation in both cell lines. Of those proteins six were up-regulated in both cell lines: CD44; Amyloid-beta precursor protein (APP); N-acetyl-alpha-glucosaminidase (NAGLU); Coronin 7 (CORO7); Anterior gradient protein 2 homolog (AGR2); and Phospholipid scramblase 1 (PLSCR1). Six were down-regulated in both cell lines: Vacuolar protein sorting-associated protein 45 (VPS45); RNA binding motif single stranded interacting protein 2 (RBMS2); Serine/threonine-protein kinase RIO1 (RIOK1); Rap1 GTPase-GDP dissociation stimulator 1 (RAP1GDS1); DNA-directed RNA polymerase III subunit RPC4 (POLR3D); and Complement decay-accelerating factor (CD55) (Table 1 and Figure 3).

### 3.3. Validation of the SILAC Proteomics Approach Using Immunodetection of CD44 and CD63

To validate the findings of the quantitative proteomics methodology, two proteins with up-regulated expression in the resistant sublines were further investigated using immunodetection techniques: CD44, which was up-regulated in both cell lines, and CD63, which was up-regulated in DLD-1/5-FU. Their roles in resistance are covered in the discussion section.

Expression of CD44 protein was further analysed using immunoblotting (Figure 4, Appendix A). Analysis of DLD-1 and HT-29 wild-type and resistant sublines showed a single band of 81 kDa molecular weight, which was most intense in the DLD-1/5-FU-resistant subline sample. Band intensities were adjusted relative to ß-actin and ratios for CD44 were observed to closely match the SILAC data (Figure 4).

Additional validation of DLD-1/5-FU proteomics results obtained by the SILAC approach (Appendix A) was carried out for CD63 protein by immunofluorescence (Figure 4F,G). CD63 expression was significantly increased in the DLD-1/5-FU-resistant subline (Figure 4F) compared to the weakly labelled DLD-1 parent cell line (Figure 4G).

## 4. Discussion

In this study, we applied a pioneering SILAC proteomics approach to identify potential novel markers for 5-FU resistance in CRC. Whilst Tam et al. recently reported proteomic analysis of CRC-resistant cell lines [26], the approach described here employs a unique strategy to identify candidates for potential novel 5-FU resistance mechanisms in drug-resistant sublines of CRC cell lines generated in house, incorporating (i) parallel generation of high-passage wild-type controls for each cell line to enable distinction between resistance and long-term culturing, (ii) a multiplexed SILAC quantitative approach for accurate comparison of expression changes between two controls (LP and HP) and the resistant cell lines, and (iii) 2D LC separation to increase proteome coverage during MS and MS/MS analysis. The results in terms of differential expression of two selected proteins with known roles in 5-FU resistance were then validated using immunodetection methods.

Employing 2D-LC-MS and SILAC in our methodology provides us with higher sensitivity and accuracy in relative quantification, allowing more in-depth analysis of the proteome and, consequently, detection of more proteins with higher confidence. In addition, this was the first study of its type to use control parental cell lines grown for the same number of passages as the resistance cell lines, which could then be used in a multi-SILAC strategy to identify potential resistance-specific molecular mechanisms. As can be seen from the large differences seen in protein expression profiles between the early- and late-passage wild-type cell lines, this is a crucial first step to take in analysis to exclude proteins which altered expression levels solely due to pressures of continued extended passaging, as opposed to pressures of prolonged 5-FU exposure. For any such studies where there is ‘omics’ analysis of prolonged drug exposure cell sublines, accounting for differences between the early- and late-passage wild-type cell line should be the first step. In concentrating on a single drug, our study aims to eliminate potential confounding factors introduced by the complexities of drug combinations. This approach allows for a more in-depth exploration of the specific mechanisms underlying chemo resistance associated with the individual drug we are investigating. This deliberate strategy not only streamlines our investigation but also facilitates a more straightforward interpretation of the results, offering valuable insights for the development of targeted therapeutic interventions.

A higher 5-FU resistance level was developed for the DLD-1 cell line compared with the HT-29 cell line (Figure 2A,B). The differences in sensitivity may be due to the loss of DNA mismatch repair (MMR) activity [27], which is characteristic of hypermutable phenotypes such as seen for the DLD-1 cell line, but not HT-29 [28]. MMR could drive somatic mutations in DLD-1 subclones, increasing development of resistant sublines (Figure 2A,B). In 2010, Sargent et al. found evidence of lack of success of 5-FU therapy in stage II and III colon cancer patients with high microsatellite instability (MSI) or defective DNA mismatch repair system [29]. Previously, similar results showed a lack of success from 5-FU therapy in CRC patients with MSI tumours [29,30,31], suggesting that 5-FU chemotherapy may be counterproductive in patients with MSI.

Validation of quantitative proteomics findings derived from a SILAC approach and statistical modelling (LIMMA analysis) was carried out using immunoblotting and immunocytochemistry techniques for the detection and visualisation of up-regulated proteins CD44 and CD63, as identified in the proteomics analysis.

CD44 protein is known to be involved in drug resistance and to act as a cancer stem cell (CSC) biomarker, as discussed below. CD63 is a protein of the tetraspanin family, also known as transmembrane 4 superfamily. Tetraspanin proteins participate in numerous physiological processes like cell adhesion, motility, and proliferation. Additionally, CD63 activity is related to proto-oncogene tyrosine-protein kinase Src (SRC), which was found to be overexpressed in both resistant cell lines during the proteomics study. SRC has been described to be one of the proto-oncogenes with more relevance during chemotherapy resistance development, including resistance to 5-FU in CRC [32]. CD63 has been suggested as a resistance mediator by tumour-derived exosomes containing doxorubicin and other cytotoxic agents in MCF7 cell lines [33] and mediates anoikis resistance in murine melanoma cell lines [34]. However, the role of CD63 in CRC is unclear although, along with α3-Integrin, it showed higher expression in human colon carcinoma cells with spontaneous metastatic ability [35]. Alterations in expression of CD63 have been observed to occur in apoptotic peripheral blood leukocytes (PBL) of CRC patients after 5-FU administration [36]. Exosomes may be acting as shed vesicles that allow CRC cells to export 5-FU or cytotoxic products and may be considered as possible mechanisms of resistance in CRC to 5-FU by decreasing harmful effects of the drug within these cells.

Whilst the proteomic analyses identified several proteins which were up-regulated or down-regulated in either the DLD-1/5-FU or HT-29/5-FU sublines (see Section 3.3), these are not considered further here, as the primary focus in this research is to identify potentially universal markers of resistance, which are either commonly up-regulated or down-regulated in both cell lines. Interesting resistance-associated candidate proteins seen in these categories are discussed here.

Of the six commonly up- and down-regulated proteins, there is already evidence linking CD44, APP, and AGR2 with drug resistance mechanisms as discussed below, although evidence is slight and not always related to 5-FU or CRC. Whilst no clear association with resistance was seen for some proteins, some (NAGLU, CORO7, and PLSCR1), as also discussed below, have been associated with oncogenic activity.

As previously noted, CD44 has been associated with chemo resistance in different types of cancer, including CRC [6,7,8]. CD44 is a transmembrane protein which acts as a receptor for hyaluronic acid, which activates intracellular survival pathways under stress conditions such as chemotherapy. It is involved in proliferation, differentiation, and motility [37]. A study in CRC patients with hepatic metastasis found the highest expression of CD44 gene by reverse transcription PCR in CRC patients with hepatic metastasis tissue, followed by non-hepatic metastasis CRC patients and normal mucosa [38]. A study of radiation resistance in DLD-1, HCT-116, and HT-29 CRC cell lines indicated that high expression of CD44 is associated with increased radiation resistance [39]. Additionally, CD44 is known to be a cell surface biomarker of cancer stem-like cells (CSLCs), and tumour cells with CSLC properties have high proliferative activity, clonogenic activity, tumour growth, and decreased apoptosis, and are able to survive chemotherapy [40,41]. Nautiyal et al. observed a significant increase of CD44 expression levels of colon mucosa cells in 20-month-old Fisher-344 rats when they were compared with young, 5-month-old rats [42]. Comparable results were found by Patel et al. in normal mucosa of patients with adenomas. The study showed an increase of CD44 levels in subjects over 55 years when they were compared to patients below 55 years, suggesting ageing increases the number of stem-like cells that may lead to a greater predisposition in colonic mucosa to subsequent transformation [43]. CD44 is crucial for maintaining the CSC phenotype and for supporting cancer cell expansion in both in vitro and in vivo colorectal cancer cell studies [44]. Hence, the presence of 5-FU in cell growth media during a high number of passages may be acting as a selection pressure factor for resistant subclones with CSC phenotype features. DLD-1 and HT-29 subpopulations with CSC features would have the capacity to self-renew and differentiate faster, with new aberrant signalling pathways leading to the development of mechanisms of resistance to 5-FU.

From the proteomics data, a number of proteins were identified in the resistant cell lines, which supported CD44’s role in enhanced stemness as a potential mechanism of resistance to be further explored. For HT-29 in particular, increased levels of stem cell proliferation marker Epcam (SILAC ln ratio, L [resistant]/H [wild type] of 1.489) and generic proliferative marker Ki-67 (0.350), along with decreased expression of proapoptopic regulators p53 (−0.552), Bax (−1.139), APAF1 (0.746), and caspase 8 (−0.495), and increased expression of Caspase activity and apoptosis inhibitor 1 (0.878), all key components of the intrinsic pathway, were seen, whilst for DLD-1, proliferative marker Ki-67 was increased (0.546), while apoptopic factors p53 (−1.215) and Bax (−0.296) showed decreasing levels.

Of the other commonly up-regulated proteins, APP is a cell surface receptor found physiologically on the neurons and interacts with APP molecules on surrounding cells to promote transsynaptic adhesion [45]. APP proteins are expressed in gastrointestinal tumours and their members are transported to the cell membrane, where they act in cell–cell interaction, cell adhesion, and metastatic processes. It has previously been shown in hepatic cancer cell lines that APP overexpression contributes to the resistance of liver cancer cells to 5-FU [46].

AGR2 is required for MUC-2 synthesis and secretion in the production of mucus by intestinal cells. It is considered as a proto-oncogene playing a role in cell growth, cell migration, and cell differentiation, including in CRC [47]. In an SW480 tumour xenograft model, knockdown of AGR2 was seen to mediate the resistance to 5-Aza-2′-deoxycytidine [48].

NAGLU catalyses degradation of heparan sulphate and has been proposed through experiments in drosophila as protection against stress resistance and neurodegeneration [49]. In addition, fusion of the NAGLU gene with the Ikaros family zinc finger protein 3 (*IKZF3*) has been shown to be oncogenic in CRC patients [50].

CORO7 promotes F-actin polymerisation and post-Golgi trafficking [51]. A study has shown that serum autoantibodies generated against CORO7 were seen in CRC patients and found to interact directly with the proto-oncogene tyrosine kinase SRC [52].

PLSCR1 plays an important role during apoptosis through recognition of damaged cells by the reticuloendothelial system [53]. Inhibition of PLSCR1 protein by an antiphospholipid scramblase 1 antibody was shown to have an anti-proliferative effect in CRC cell lines in vitro. Additionally, PLSCR1 is thought to be involved in tumour proliferation because its overexpression has been observed as part of EGF stimulation pathways. The enzyme has been seen to be overexpressed in CRC patient samples and linked to poor prognosis [54]. Expression has also been linked to therapy-resistant glioblastoma multiforme patients [55].

Of the six commonly down-regulated proteins, some evidence could be found to link all, apart from POLR3D, with involvement in drug resistance mechanisms as discussed below.

VPS45 plays a role in vesicle-mediated protein trafficking from the Golgi stack through the trans-Golgi network [56]. Increased sensitivity to methylmercury with down-regulation of endosome trans-Golgi transport intensified the resistance phenotype in a yeast model [57].

RBMS2 has been implicated in DNA replication, gene transcription, cell cycle progression, and apoptosis [58]. Xu et al. demonstrated that overexpression of RBMS2 in breast cancer cells increased their sensitivity to doxorubicin [59].

RIOK1 participates in maturation of the 40S ribosomal subunit [60]. It has been shown, in the same study, to promote tumour growth in a CRC cell line and to be tumourigenic in a lung cancer xenograft model [61]. Knockdown of RIOK1 has also been found to mediate radio resistance in CRC cells [62].

RAP1GDS1 participates in activation of G proteins like Rap1a/Rap1b, RhoA, RhoB, and KRAS, and promotes the GDP/GTP exchange reaction by stimulating GDP dissociation from G proteins [63]. A study in prostate cancer cells demonstrated high expression of RAP1GDS1 in cells under hypoxia and postulated that targeting it and similar CLK kinases may provide benefit in the treatment of cancers in which tumour hypoxia contributes to resistance to therapy [64].

CD55 is a major regulator of both the classical and alternative complement activation pathways [65]. CD55 overexpression has been linked to aggressiveness and poor prognosis [66], and antibody-targeting controlled CRC xenograft growth in mice [67].

In terms of proteins which are known to be typically involved in the development of multidrug resistance, then, some changes were seen, although not consistently for both cell lines. Markers of drug resistance, ALDH1A3 (5.352), ABCC1 (1.803), and ABCG2 (1.569) were significantly escalated in HT 29, with ABCB1 (0.639) and ABCG2 (1.840) escalated with DLD-1. NF-κB [68] was seen to be up-regulated in DLD-1 (1.010), with no change in HT-29, whilst thymidylate synthase was not detected at the level utilised in this proteomic strategy.

As discussed above, for 11 out of the 12 commonly identified up-regulated and down-regulated proteins there is already some evidence for association with drug resistance mechanisms, or role in CRC oncogenesis. This suggests that there is value in progressing to mechanistic studies using the resistant cell line panel we have developed here, as well as validation in archival clinical material from patients treated with 5-FU-containing therapy where sensitivity or resistance to the regime is known.

## 5. Conclusions

In conclusion, we have demonstrated the power of applying a SILAC proteomic approach and the quantification of protein expression to identify potential markers for 5-FU drug resistance by comparing protein expression between 5-FU-resistant sublines and wild-type sensitive parental cell lines with high and low passaging. The importance to this methodology of using both low- and high-passage wild-type cell lines, and of validation using conventional immunodetection techniques has been highlighted.

We have revealed a panel of commonly altered proteins during development of acquired resistance to 5-FU, and these findings will act as a source of potential target proteins for extensive mechanistic studies going forward.

## Figures and Tables

**Figure 1 cells-13-00342-f001:**
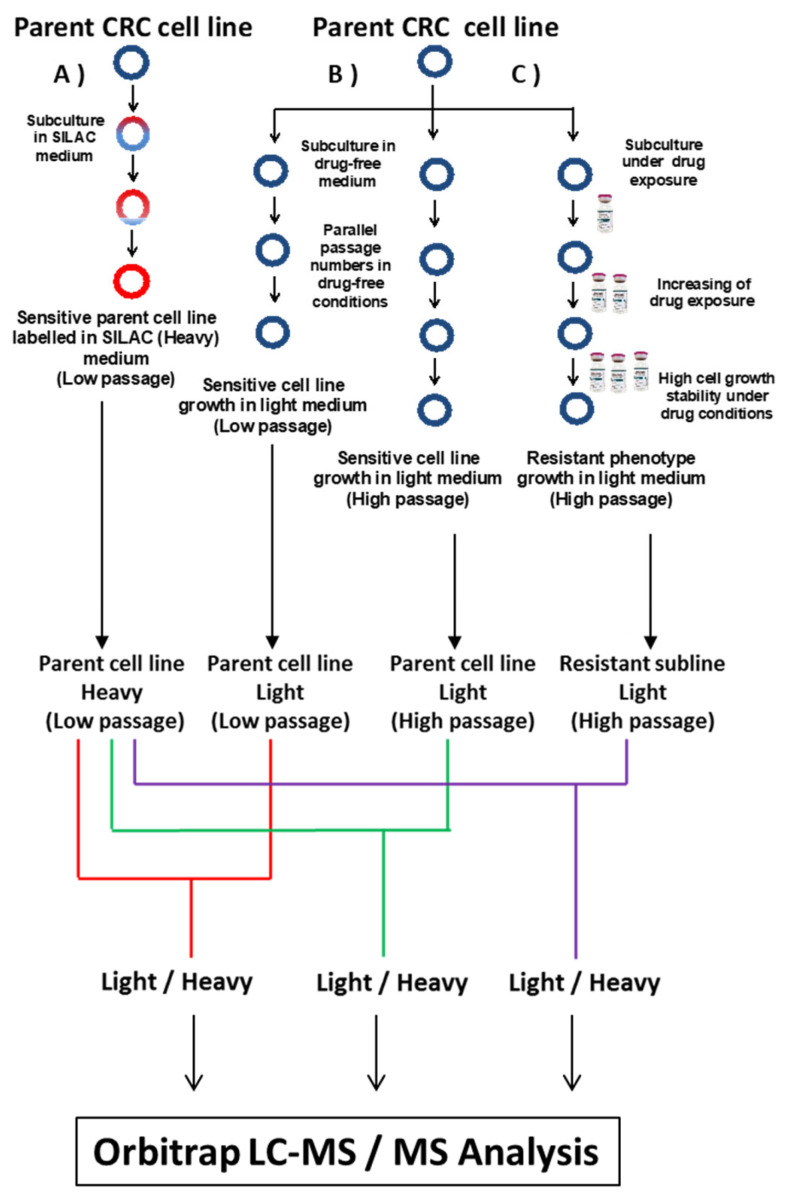
Flowchart for the SILAC experimental approach showing (**A**) parent cell line growth in SILAC medium with Heavy amino acids that are incorporated to parent cell line during nine passages. (**B**) Parent cell line growth in drug-free RPMI medium to be used as a control during protein quantification, and (**C**) parent cell line growth in 5-FU-containing RPMI medium during the process of the establishment of CRC-resistant sublines. The strategy was applied in parallel for DLD-1 and HT-29 cell lines.

**Figure 2 cells-13-00342-f002:**
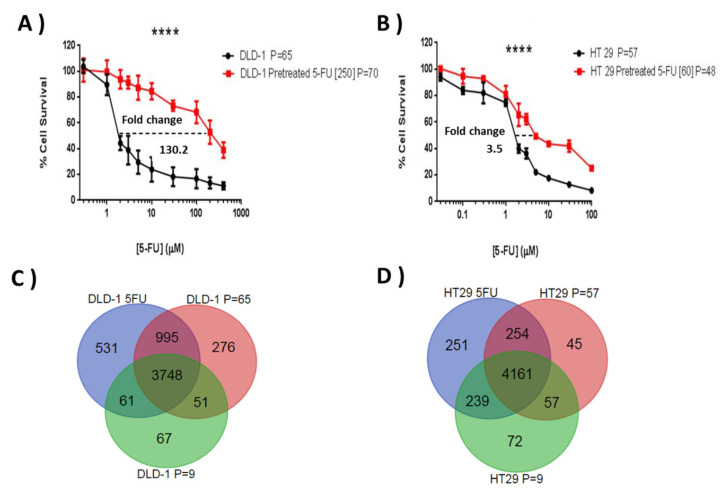
Comparative 5-FU sensitivity of the generated resistant sublines and their respective wild types (**A**,**B**), and the similarities and differences seen in terms of protein expression for the different sublines (**C**,**D**). Graphs (**A**,**B**) show cell survival profiles of parent cell lines and their respective resistant sublines to 5-FU for (**A**) DLD-1 (**B**) HT-29 CRC cell lines under exposure to (**A**) 5-FU [1–1000 µM] and (**B**) 5-FU [0.03–100 µM] doses over 96 h. MTT assays were performed in three independent experiments. The difference in IC_50_ among parental cell line and resistant 5-FU sublines is highly significant in the three experiments (*p* ≤ 0.0001). A 130.2-fold change and 3.5-fold change difference in IC_50_ was found in the DLD-1/5-FU [250 µM] and HT-29/5-FU [60 µM] resistant sublines, respectively (**** *p* ≤ 0.0001). The (**C**,**D**) Venn diagrams show the number of proteins commonly quantified in the two parent cell lines (**C**) DLD-1 and (**D**) HT-29, with a high and a low number of passages and in their respective resistant sublines to 5-FU. P refers to the number of cell passages carried out when the subline was assessed.

**Figure 3 cells-13-00342-f003:**
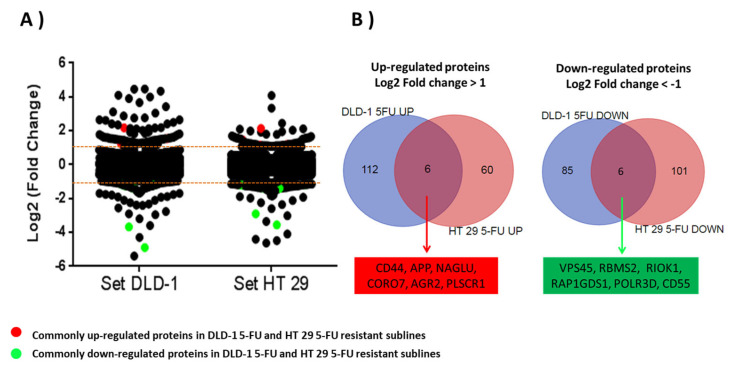
(**A**) Scatter plot figure showing dispersion of log_2_-fold change ratios for 3003 proteins commonly quantified in parent cell lines and 5-FU-resistant sublines. All proteins with a log_2_-fold change higher than ±1 were considered as altered proteins in resistant sublines. Commonly up-regulated (red) and down-regulated (green) proteins in both 5-FU-resistant sublines are highlighted. (**B**) Venn diagrams showing significantly up-regulated and down-regulated proteins in resistant sublines DLD-1/5-FU and HT-29/5-FU.

**Figure 4 cells-13-00342-f004:**
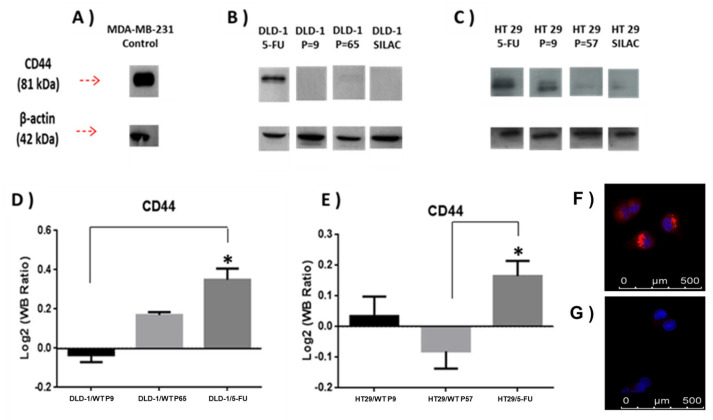
(**A**) Densitometry for CD44 expression was performed for (**B**) DLD-1 and (**C**) HT-29 parent cell lines and 5-FU-resistant sublines and analysed using a *t*-test. (**D**,**E**) show that relative expression of CD44 abundance in the indicated samples was divided among CD44 abundance in SILAC sample, and relative abundance was calculated as previously during the SILAC protocol to estimate SILAC ratios (L/H). SILAC ratios (L/H) for CD44 were DLD-1/5-FU (4.7), DLD-1 P = 9 (0.44), DLD-1 P = 65 (1.2), HT-29/5-FU (7.01), HT-29 P = 9 (2.8), and HT-29 P = 57 (0.42). Data are presented as mean (n = X); an asterisk represents significant differences (Student’s *t*-test, * *p* < 0 0.05). MDA-MB-231 mammary adenocarcinoma cell line was used as a positive control cell line for high CD44 expression. (**F**,**G**) show immunolocalization of CD63; in (**F**) DLD-1/5-FU-resistant subline with strong labelling, and in (**G**) DLD-1 parent cell line with weak labelling.

**Table 1 cells-13-00342-t001:** Characteristics of the proteins commonly up-regulated and down-regulated in the 5-FU-resistant DLD-1 and HT-29 cell lines.

	Gene Name	UniProt Accession ^1^	MW ^2^ [kDa]	Calc. pI ^2^	Seq Coverage ^3^	Peptides ^4^	PSMs ^5^	Mascot Score ^6^	Abundance Ratio: Log_2_ R/LP ^7^	
Commonly up-regulated									DLD-1	HT-29
CD44 antigen	CD44	P16070	81.5	5.3	8.2	5	76	1261	0.97	2.23
Amyloid-beta A4 precursor protein	APP	P05067	86.9	4.8	24.8	14	103	1122	1.39	1.04
N-acetyl-alpha-glucosaminidase	NAGLU	P54802	82.2	6.7	26.9	12	61	561	1.45	1.43
Coronin 7	CORO7	P57737	100.5	5.8	28.6	12	64	864	1.09	0.82
Anterior gradient protein 2 homolog	AGR2	O95994	20.0	9.0	57.1	10	488	7943	0.32	2.47
Phospholipid scramblase 1	PLSCR1	O15162	35.0	4.9	9.4	3	27	515	0.74	1.30
Commonly down-regulated										
Vacuolar protein sorting-associated protein 45	VPS45	Q9NRW7	65.0	8.2	11.9	6	30	377	−4.83	−2.93
RNA binding motif single stranded interacting protein 2	RBMS2	Q15434	43.9	9.1	27.5	6	27	533	−0.87	−2.00
Serine/threonine-protein kinase RIO1	RIOK1	Q9BRS2	65.5	6.2	13.7	4	11	150	−1.13	−0.05
Rap1 GTPase-GDP dissociation stimulator 1	RAP1GSD1	P52306	66.3	5.3	23.9	10	50	1010	−1.75	−1.46
DNA-directed RNA polymerase III subunit RPC4	POLR3D	P05423	44.4	7.0	4.5	1	7	159	−0.33	−0.09
Complement decay-accelerating factor	CD55	P08174	41.4	7.6	25.5	5	15	123	−2.28	−1.78

^1^ Accession number is the unique identifier in the UniProt database (https://www.uniprot.org/, accessed on 1 September 2017); ^2^ molecular weight and pI are theoretical values based on the amino acid sequence; ^3^ sequence coverage is the percentage of the amino acid sequenced covered by the ^4^ peptides unique to the protein which were identified from the total number of ^5^ peptide spectral matches. ^6^ Mascot Score is the database search engine confidence score for the protein identity where *p* < 0.05 significance is achieved for a value of 28 or higher (https://www.matrixscience.com/, accessed on 1 September 2017). ^7^ The abundance ratios are the expression levels determined by MS analysis of the SILAC ratios, where R is the value for Light isotope data in the resistant cell lines and LP is the value of the Heavy isotope data in the Heavy-labelled low passage control.

## Data Availability

The mass spectrometry proteomics data have been deposited 10 December 2023 to the ProteomeXchange Consortium via the PRIDE (https://www.proteomexchange.org/) partner repository with the dataset identifier PXD047672.

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
