# Peer review of "A Proteomic Investigation to Discover Candidate Proteins Involved in Novel Mechanisms of 5-Fluorouracil Resistance in Colorectal Cancer"

_cells, 2024, doi:10.3390/cells13040342_

Round 1

Reviewer 1 Report

Comments and Suggestions for Authors

Overall, the manuscript is too preliminary to be considered for publication.

This manuscript offers limited conceptual advancement in our understanding of 5-FU resistance mechanisms. The primary weakness of this manuscript is the lack of mechanistic data. More mechanistic insights into the role of CD44 in 5-FU resistance mechanisms are required.

Comments on the Quality of English Language

Moderate editing of English language required.

Author Response

Comments and Suggestions for Authors

Overall, the manuscript is too preliminary to be considered for publication.

This manuscript offers limited conceptual advancement in our understanding of 5-FU resistance mechanisms. The primary weakness of this manuscript is the lack of mechanistic data.

We thank the reviewer for their feedback. In response to this, we have revised the title to better reflect the nature of our investigation as an extensive proteomics-based study aimed at identifying potential candidate proteins for further exploration of their roles in 5-FU resistance mechanisms, which is beyond the remit of this study. Furthermore, we have made corresponding adjustments in the text to emphasize the exploratory nature of our approach.

In addressing the mechanistic aspects of 5-FU resistance, our manuscript incorporates significant insights through Gene Ontology analysis, detailed in the Results section 3.2, as well as in the Discussion section. These sections elucidate essential biological processes relevant to cell resistance, including DNA repair, transcription activity, and apoptosis, specifically within the context of both colon cancer cell lines under investigation.

The primary objective of our study was the identification of potential candidate proteins associated with 5-FU resistance mechanisms. In line with this focus, we have concentrated our efforts on individual protein candidates, leveraging the informative data obtained through shotgun proteomics.

To enhance the manuscript's quality, we have performed a comprehensive Functional Enrichment analysis using the FunRich tool, facilitating the integration of Gene Ontology data seamlessly into both the Results and Discussion sections. Typically, FunRich incorporates widely used biological databases such as Gene Ontology (GO), Kyoto Encyclopedia of Genes and Genomes (KEGG), Reactome, Uniprot, Human Protein Atlas and other relevant repositories.

More mechanistic insights into the role of CD44 in 5-FU resistance mechanisms are required.

We have expanded our discussion section on CD44 to give some further evidence that supports the role of CD44 in cancer stemness as a rationale for its involvement in 5-FU resistance.

Comments on the Quality of English Language

Moderate editing of English language required.

We have extensively reviewed and proof-read the revised manuscript and hopefully addressed any potential issues with the English language. While we believe significant improvements have been made, unfortunately as specific examples highlighting inadequacies were not provided, it makes it challenging for us to pinpoint the exact areas that need adjustment, but hopefully we have addressed your concerns.

Reviewer 2 Report

Comments and Suggestions for Authors

This manuscript reports on a SILAC proteomic approach to quantify the protein expression in 5-FU resistant sublines compared to WT sensitive parental cell lines to identify potential markers for 5-FU MDR.

The manuscript provides the reader with useful information about the role of the twelve proteins involved. The manuscript has been well organized and is easy to follow. 

The provided scientific information may be useful to scientists working in the field of MDR drug resistance.

I feel, the manuscript may be considered for acceptance after minor changes.

Minor points

The authors decided to use DLD-1 HT-29, and MDA-MB-231, as well as L- Proline and L-Arginine amino acids. An explanation should be given.

Figure 2, panel C and D, I assume that P means the number of passages. It should be explained in figure legend. 

Figure 4, panels D and E, abscissa values are not easy to read.

Author Response

Comments and Suggestions for Authors

This manuscript reports on a SILAC proteomic approach to quantify the protein expression in 5-FU resistant sublines compared to WT sensitive parental cell lines to identify potential markers for 5-FU MDR.

The manuscript provides the reader with useful information about the role of the twelve proteins involved. The manuscript has been well organized and is easy to follow. 

The provided scientific information may be useful to scientists working in the field of MDR drug resistance.

I feel, the manuscript may be considered for acceptance after minor changes.

 We thank the reviewer for their constructive and helpful feedback and have addressed the revisions they have suggested as covered below:

Minor points

 The authors decided to use DLD-1 HT-29, and MDA-MB-231, as well as L- Proline and L-Arginine amino acids. An explanation should be given.

Justification for the choice of cell lines and amino acids has now been added to the Materials section 2.1. This hopefully provides clarity on the rationale behind our experimental choices.

Figure 2, panel C and D, I assume that P means the number of passages. It should be explained in figure legend. 

Yes it does - a comment has been added to the figure legend to clarify this

 Figure 4, panels D and E, abscissa values are not easy to read.

The charts have been revised to improve the legibility of the abscissa

Reviewer 3 Report

Comments and Suggestions for Authors

Introduction – CD44 and CD63 – this molecules are cell surface or vesicle surface molecules not directly associated with chemotherapy resistance (like MDR1 for instance) – should be pointed more precise, or said that or said that, among others can be associated with resistance.

Results:

The results/discussion section lacks reference to ABCC1, ABCB2 typically involved in the development of multidrug resistance including 5FU resistance. Even if the expression of these proteins has not changed, it is worth mentioning it to have a broader reference.

What about thymidylate synthase, or NF-kappa B?

Overall it is interesting work – demonstrating methods for studying the resistance of colon cancer cells.

However, there are a few issues in the discussion that could improve the significance of this manuscript. First of all, it is a study on two cell lines. There are studies on clinical material on genes and proteins whose expression changes under the influence of 5FU. There are also numerous established mechanisms of resistance to 5FU - how does this relate to these two proteins CD44 and CD63, the change of which is reported by the authors of this manuscript?

Author Response

Comments and Suggestions for Authors

We thank the reviewer for their constructive and helpful feedback and have addressed the revisions they have suggested as covered below:

Introduction – CD44 and CD63 – this molecules are cell surface or vesicle surface molecules not directly associated with chemotherapy resistance (like MDR1 for instance) – should be pointed more precise, or said that or said that, among others can be associated with resistance.

 We have adjusted the introduction to include MDR1 as an example of a known mechanism of resistance for 5-FU, and highlighted the possibility of other cell/vesicle surface markers such as CD44 also potentially having a role in resistance.

Results:

The results/discussion section lacks reference to ABCC1, ABCB2 typically involved in the development of multidrug resistance including 5FU resistance. Even if the expression of these proteins has not changed, it is worth mentioning it to have a broader reference.

What about thymidylate synthase, or NF-kappa B?

We have included a paragraph in the discussion section that highlights our observations for such proteins, and given reason for why they did not come up as top candidates in our study. The SILAC strategy yielded above average results in detecting approximately 50% of the total proteome (4000 to 5000 proteins) for each cell line, with particularly a number of known drug-resistance components quantified, although thymidylate synthase was not detected.

Overall it is interesting work – demonstrating methods for studying the resistance of colon cancer cells.

 However, there are a few issues in the discussion that could improve the significance of this manuscript. First of all, it is a study on two cell lines. There are studies on clinical material on genes and proteins whose expression changes under the influence of 5FU. There are also numerous established mechanisms of resistance to 5FU - how does this relate to these two proteins CD44 and CD63, the change of which is reported by the authors of this manuscript?

The use of cell lines instead of tissues gives us a purer sample of a proteome where the only variable is drug resistance. This is as opposed to tissues where the complex cellular heterogeneity and stromal composition greatly dilutes the cancer cell proteome of interest. Thus, any changes in expression which are seen in our samples are solely down to alterations due to drug resistance and gives us a better indication of potential proteins involved in resistance. In addition, the SILAC technique cannot be performed on tissues or biopsies as it is not possible to generate a labelled control.

The feedback focuses on CD44 and CD63, yet these are just a few of the proteins that we have highlighted in our study as being of potential importance in 5-FU resistance - they are perhaps seen as more prominent due to their use as markers to validate our findings. Whilst we have expanded the discussion to further cover the potential role of CD44 and how it relates to other established mechanisms of resistance as a marker for stem-ness, and also give evidence for how CD63 may be involved in resistance, the aim of this study was to try and identify markers which were not definitively related to established mechanisms of resistance, but rather could potentially be involved in novel mechanisms.

Round 2

Reviewer 1 Report

Comments and Suggestions for Authors

I am satisfied with the revisions that have been made by the authors.